# The Effect of Aquatic Exercise on Postural Mobility of Healthy Older Adults with Endomorphic Somatotype

**DOI:** 10.3390/ijerph16224387

**Published:** 2019-11-10

**Authors:** Khadijeh Irandoust, Morteza Taheri, Masoud Mirmoezzi, Cyrine H’mida, Hamdi Chtourou, Khaled Trabelsi, Achraf Ammar, Pantelis Theodoros Nikolaidis, Thomas Rosemann, Beat Knechtle

**Affiliations:** 1Department of Soirt Sciences, Faculty of Social Sciences, Imam Khomeini International University, Qazvin 34148-96818, Iran; irandoust@soc.ikiu.ac.ir (K.I.); m.taheri@soc.ikiu.ac.ir (M.T.); 2Faculty of Physical Education and Sport Science, Islamic Azad University, Tehran 1439813117, Iran; massoudmirmoezi@live.com; 3Education, Motricité, Sport et Santé (EM2S), High Institute of Sport and Physical Education, University of Sfax, Sfax 3000, Tunisia; sirinehmida@hotmail.fr (C.H.); trabelsikhaled@gmail.com (K.T.); 4Institut Supérieur du Sport et de l’éducation physique de Sfax, Université de Sfax, Sfax 3000, Tunisia; h_chtourou@yahoo.fr; 5Activité Physique, Sport et Santé, UR18JS01, Observatoire National du Sport, Tunis 1003, Tunisie; 6Institute of Sport Science, Otto-von-Guericke-University, 39104 Magdeburg, Germany; ammar.achraf@ymail.com; 7Exercise Physiology Laboratory, 18450 Nikaia, Greece; pademil@hotmail.com; 8Institute of Primary Care, University of Zurich, 8091 Zurich, Switzerland; thomas.rosemann@usz.ch; 9Medbase St. Gallen Am Vadianplatz, 9001 St. Gallen, Switzerland

**Keywords:** water exercise, postural mobility, fall, endomorph

## Abstract

The fear of falling (FOF) limits the movements of the older adults, which, in turn, might impair postural mobility. An aquatic environment has a relatively low risk of falling and can improve motor abilities. The aim of this study was to investigate the effect of aquatic exercise on postural mobility of the healthy endomorph elderly somatotype. Therefore, 37 healthy endomorphic older adults with an average age of 64.38 ± 4.12 years participated in this study. Participants were randomly divided into four groups (i.e., Aquatic exercise, Dry-land exercise, Aquatic control, and Dry-land control). The Heath-Carter method was used to estimate the criterion somatotype, and the Tinetti method was used to determine postural mobility. Covariance analysis was used to examine the mean differences at a significance level of *p* < 0.05. The results showed that there was a significant difference between the aquatic exercise group and the two control groups (*p* < 0.01), and the dry-land exercise group was significantly different from the aquatic control (*p* < 0.05) and dry-land control groups (*p* < 0.01). The results indicate that the design of aquatic exercise programs, especially for endomorphic older adults with inappropriate body shape, for whom dry-land exercises are not appropriate, likely, has a positive effect on the motor control and both the balance and gait and provide appropriate postural mobility without FOF in older adults.

## 1. Introduction

Anatomical differences regarding physical and motor performance of individuals are not easily identifiable and require experimental study and research. These differences can affect people’s general health [1,2]. One of the most important goals of general health is to reduce age-related disabilities in older adults [3].

In this regard, appropriate physical activity of the elderly is one of the methods that can be used to prevent, postpone, or treat problems caused by the aging process [4]. Falling is one of the major concerns among older adults. Its causes include reduced strength, flexibility, muscular endurance, body type, problems with vision, hearing, vestibular system, diet, and fear of falling, all of which change the walking patterns and even postural mobility [5,6].

Balance is a complex motor skill that demonstrates the postural mobility in preventing falling [7]. The ability to maintain body’s balance is an essential factor in performing daily activities. Older adults try to reduce the risk of falls by limiting their movement and activities so that they would increase balance and postural mobility [8]. By limiting these activities, the elderly are trapped in an vicious cycle, which includes voluntary immobilization, motor impairment, loss of postural skills, instability to stand, and Fear Of Falling (FOF) [9].

Voluntary immobilization, movement diagnosis, and body adaptations to the surrounding environment all result in both reduced physical activity and involuntary postural mobility. They also increase the postural instability and the FOF. This cycle frequently happens until it makes the older adults severely homebound [7,9,10].

Determinants of balance include age, gender, and some anthropometric features (e.g., height, weight, limb length, limb circumference, limb width, subcutaneous fat, body type) [11]. Morphological or body type is one of the factors affecting balance [12]. Body composition is affected by the daily physical activities and vice-versa [4]. The three different body types are ectomorph, endomorph (which expresses body fat content), and mesomorph [13]. As abdominal obesity increases, a compensatory arch is created in the lumbar curve with the increase of the lumbar arch, the center of gravity shifts forward, causing changes in walking, postural control disorders, more falling, and reduced motor mobility [14]. Researchers have mentioned that the length of step, Range of Motion (ROM) and balance are affected by body type [13,15,16]. In a study conducted by Cao et al. it was mentioned that regular physical activity and exercise can effectively decrease the risk of falling and promote the mobility and safety movements in the old women [17]. 

Since one of the major goals of physical education and sport sciences is the provision of a sound life as well as a healthy society, studies concerning the health of older adults seem essential. Therefore, it is important for the community to find ways that contribute to improve healthy life of this population [18]. The area of mobility in the older adults is one of the key issues that are noticeable in young [2] and older adults’ well-being. Degradation changes in the physical and psychological potentials of the body caused by the environment affect the older adults’ lifestyle [19] and ultimately lead to FOF, which disrupts their daily performance and postural mobility. Symptoms such as pain, muscle weakness, lack of balance, obesity, arthritic diseases, walking disorders, FOF, makes exercising on the ground for the older adults far more difficult [8,20,21]. Due to FOF and injuries happening during traditional exercises on land, this type of exercise is not suitable for all older adults, especially those who are overweight and obese [9]. However, the aquatic environment because of favorable mechanical properties of water on the body (e.g., reducing the pressure on the joints, reducing the heart rate, improving the balance, and less occurrence of physical damage in water and lower risk of injury), it seems appropriate for older adults with FOF [22]. Taheri et al. argued that exercise training in water improves muscular balance and muscle strength and is the most important indicator of preventing the risk of falling in the older adults [10]. Favorable mechanical properties of water on the body are: reducing the pressure on the joints, reducing the heart rate, improving the balance, and less occurrence of physical damage in water.

It seems that by practicing postural mobility exercises in a safe environment, such as the aquatic environment and a proper exercise plan that takes into account the physiological, physical, and motor constraints, older adults can somewhat compensate for decreased mobility, FOF, and postural mobility. Due to the paucity of research in this area and the lack of a suitable exercise plan, this study was conducted to investigate the effect of aquatic exercises on the postural mobility of the healthy older adults with endomorphic somatotype.

## 2. Materials and Methods

### 2.1. Ethical Approval

The research was approved by the local ethical committee of Imam Khomeini International University (ref. no: 18145).

### 2.2. Subjects

The present study was a pre-/post-test experimental design with a control group. Due to availability, a sample of 65 older adults who led an independent life (i.e., sample size suggested by Byrne et al. and 80% power efficiency to detect group differences at *p* < 0.05) [23] from the city of Qazvin (30 men and 35 women, 65 ± 4.9 years, ages 60–70 years) volunteered to participate to the study. For observing the ethics, besides getting consent from all subjects, it was explained that the results of the study were purely for research purposes and would be published collectively without mentioning the names of the individuals. Their participation in the study was also optional, and they could leave it at any stage.

The inclusion criteria included over 60 years, BMI over 30 kg/m^2^, waist-to-hip Ratio (WHR) above 0.9 in male and 0.8 in females [24], lack of regular exercise or physical activity in the past six months, no vision problems, hearing and vestibular functioning problems, not taking tranquilizer and antidepressants that affect balance, and no history of lower limb injury over the last year. Subjects, also, conducted a Short Physical Performance Battery Protocol (SPPB) and those who scored a score of less than nine (SPPB ≤ 9) were excluded from the study. A total of 25 older adults were excluded from the research due to lack of inclusion criteria.

Participants were randomly divided into four groups: Aquatic Exercises (AE) (4 males, 6 females), Dry-land Exercises (DE) (5 males, 5 females), Aquatic Control (AC) (5 males, 4 females), and Dry-land Control (DC) (3 males, 5 females). It is worth noting that two participants from the DC group and one from the AC group discontinued the treatment. The reasons given were changes in work schedule or home responsibilities. Eighty-seven percent of the 23 exercise subjects (1 with AE and 2 with DE) who began classes finished the eight-week courses and were retested.

Prior to the beginning of the exercising period, a medical history questionnaire and a Physical Activity Readiness Questionnaire (PAR-Q) were used to determine the health status and level of readiness of the participants. All of the participants in the exercise group were qualified to enter the exercise program. The PAR-Q questionnaire is a screening and qualitative assessment tool for participating in physical exercises [8]. PAR-Q is recommended by the committee to be used as a standard for entering the exercising programs of moderate intensity [14].

The Heath-Carter method was used to determine the body types. The Heath-Carter method consists of 10 anthropometric measurements including height, weight, arm width (i.e., the width of the humerus in the elbow area), hip width (i.e., the femur width in the knee), maximum arm circumference, maximum calf circumference, subcutaneous fat in triceps, subscapularis muscle, supraspinatus muscle, and gastrocnemius muscle of individual participants. In order to measure the level of subcutaneous fat, a digital caliper (Mitutoyo, Japan) with a precision of 0.1 mm and a measure tape were used to measure the calipers subcutaneous fat (SEAHAN, model SH5020 made in Korea, with a precision of 0.5 mm) on the right side of the body. The Heath-Carter method gives three endomorphic, mesomorphic, and ectomorphic scores to each individual, and the one with one and a half points higher than other scores, considered being the body type of that person (see Table 1).

Postural mobility of the older adults was investigated by the help of the standard instrument for the Tinetti Performance Oriented Mobility Assessment (POMA). This tool is a performance test of balance and gait maneuvers used during normal daily activities. This test has two subscales of balance and gait. There are 13 maneuvers in the balance portion and 9 maneuvers in the gait portion. The total score of the two sections is 28 points. A score less than 19 out of 28 has a sensitivity of 68% and a specificity of 88% for predicting an individual who will have two or more falls [25]. This test has acceptable validity and reliability [25,26].

The four mentioned groups were composed of two exercising groups and two control groups. The exercising groups performed the protocol provided in Table 2 for eight weeks, three sessions per week for 50 min. Participants completed the first 4 weeks of exercise with a heart rate of 50% of the Maximum Heart Rate (MHR) and the last four weeks with 60% of MHR. Target Heart Rate (THR) zone was determined using the Karvonen Formula [1]. The water temperature was 29–30 °C and the water depth was 0.7 to 1.3 m [22]. A sports specialist, with a lifeguard on alert around the water to watch the subjects, conducted the training course in water. All subjects were allowed to perform exercises in water. The exercising group practices were administered on land in an enclosed sports arena at a temperature of 23–22 °C and this group was trained by a physical fitness instructor with the same heart rate intensity in the AE group. The AC group sat by the poolside that was built into the water and talked to each other, and the DC group on land also sat down on chairs near the gym and talked to each other [9].

One week prior to the main test, the anthropometric characteristics of participants in the experimental group such as age, height, weight, body mass index (BMI), body fat percentage, and body type were recorded. The height and weight of the subjects were recorded using the BSR 85 model, which includes electronic scale and mechanical height gauge. To measure the percent of body fat (PBF), a three-point subcutaneous fat (i.e., chest, abdomen, and thigh) measurement was used and then the PBF was estimated by Brozek formula [24]. To increase the validity of the test results and in order to reduce the bias, the test was administered to the same participants who were unaware of the classification in the experimental group. All the aforementioned factors except age and height were measured in the pre-test and after eight weeks in the post-test among four groups.

It is worth noting that pre-exercise conditions (i.e., from the environmental and psychological point of view) were taken into account in the post-test. Aquatic Heart Rate Deduction (AHRD) was used to more accurately estimate heart rate of the older adults in the water. AHRD is achieved by the heart rate on land minus the heart rate in water, which is less than 17 beats per min on land and should be taken into account in the Karvonen Formula [10]. The intensity of exercise recommended for older adults is based on the Aquatic Exercise Association (AEA) recommendation, that is, 90–145 beats per min [22]. Accordingly, the water sports instructor performed this moderate intensity, frequency, and speed for all training sessions.

### 2.3. Statistical Analysis

Data were checked for normality distribution. To detect differences between the groups, a one-way analysis of variance (ANOVA) and an analysis of covariance (ANCOVA) were used at the significance level of *p* ≤ 0.05.

## 3. Results

Given the values obtained, kurtosis and skewness for the variables were between −1 and +1, which showed the normal distribution of the data. Hence, parametric methods were used for data analysis. The results of the one-way ANOVA before exercise intervention showed no significant differences between any of the research variables (*p* > 0.05) supporting the homogeneity of the subjects in the corresponding randomized groups. The individual characteristics of the participants along with the results of the ANOVA test have been presented in Table 3 for the homogeneity of the groups.

The total score of postural mobility is shown in the four groups according to Tinetti test, which is a combination of the balance test (16 points) and the test of standing and walking (12 points), scoring a total of 28 points (Figure 1). The results of covariance analysis for Tinetti test (postural mobility) showed a significant difference in the four groups (AE and DE, AC and DC) (*p* < 0.001, and F_32, 3_ = 25.38) (see Table 4). The post hoc test for comparison of the means showed a significant difference between the AE group and the AC and DC groups (*p* < 0.01) and a significant difference between DE group with the AC (*p* < 0.05) and DC groups (*p* < 0.01) (Figure 2).

## 4. Discussion

The purpose of this study was to investigate the effect of aquatic exercise on the postural mobility of healthy older adults with endomorphic somatotype. The results showed that aquatic exercise can reduce the risk of falling in the older adults by increasing postural mobility through the implementation of low dynamic movements, based on dynamic flexibility, muscular and balance strength, and resistance exercises. After the training period, the results showed that the aquatic exercise group was upgraded from moderate risk of a falling to low risk. Taheri et al. stated that aquatic exercise program improves balance and muscle strength as the most important indicator for preventing the older adults risk of falling [10]. Performing movements in water due to better detection and recognition of movement error and better ROM causes that older adults receive appropriate feedback without FOF and the body’s control system becomes more efficient in the future [9].

FOF limits daily activity in older adults, which in turn reduces physical performance and social independence. The ability to maintain functional balance and postural mobility is significant and prevents falling [27]. Increased age and decreased muscle strength, balance, flexibility, and stride patterns are connected to an increased risk of falling in the older adults [28]. Thus, to prevent falling in the older adults, it is necessary to create specific conditions and a safe environment, activate the muscles involved in postural mobility, stimulate the body balance factors, and create or promote a new control system with environmental compatibility [10].

In the present study, more postural mobility was seen in the DE compared to AC and DC groups. As age increases, morphological and metabolic changes in the skeletal muscle occur in the form of a decrease in the transverse area of the muscle, which reduces the ability and weakens the function [29]. Here, the role of muscular strength is fundamental and important, otherwise according to the British Geriatrics Society (BGS) and the American Academy of Orthopedic Surgeons (AAOS), muscle weakness, reduced balance, and physical fitness increase the risk of falling by 4 to 5 times. Reduced muscle mass, muscle strength, and muscle aging, known as age-related muscle mass, can interfere with walking, disability, and falling [30]. Carmeli et al. argued that land training programs have a positive effect on the performance and mobility of the older adults [31]. However, due to FOF and injury, traditional land training programs are not suitable for all older adults with endomorphic somatotype, especially overweight and obese older adults [9].

In the literature, endomorphic people with endomorphic body type are likely to have a shorter life expectancy and the risk of falling increases [32,33]. Obese people are 20% or more heavier than normal body weight [34,35]. Aquatic exercise is suitable for obese people with osteoarthritis, low back pain, knee pain (e.g., inadequate body type and low confidence/high blood pressure and high heart rate, and FOF as in the aquatic environment). Due to lower temperature compared to the body temperature and hydrostatic pressure on the body’s heart rate, inflammation and pressure on the joints decrease, and blood circulation, endurance, and endurance of the body will increase against fatigue. The cooling effect of the water makes it easier for the older adults to run, and for the older adults with endomorphic body type. Being covered with water and hiding from the other participants, they feel less embarrassed during exercise, and adhere more to the continued activity [22].

The results showed that aquatic exercise greatly improves postural mobility, and even the AC group in the older adults has also partially experienced an increase in postural mobility. According to a source in AEA (2017), older adults at risk of falling should consider including balance exercises in their daily routine and it has been argued that physical activity in water helps the older adults to maintain their sensory, physical, and physiological functions and even improve them [22]. Mirmoezzi et al. [8,36] stated that aerobic exercise on land helps older adults to enhance balance and postural mobility and resistance training in the water are associated with reduced indices of fatigue, followed by more postural mobility and reduced risk of falling [8,36]. The viscosity and resistance of the water presents a unique challenge, by which motion of the body or its parts will be affected by the water’s viscosity and buoyancy, and somewhat by gravity, and due to this, when an individual is in water up to their waist, about 50% of their weight is lost. Besides reducing gravity and weight loss in water, the probability of falling in the water is decreased [22]. This makes the older adults experience more mobility in water with more ROM. Thus, they experience more variation with errors and more motion experience [9,22]. Moreover, the results showed that aquatic exercise improved balance due to neural adaptations induced by exercise in water, such as the use of neural efficient reorganization of cortical somatosensory, increased the efficiency and strength of synaptic connections, increased activation of the nervous system, reduced neural inhibitory responses, decreased the resistance of neural pathways to impulse transmission, and improved and facilitated the transmission of inputs of each sensation [27,37]. Most researchers have considered that these positive effects of exercise in elderly will help maintain good blood supply in many organs, especially the central nervous system and peripheral. This helps keep the balance in control, further improving processing efficiency and postural mobility [38,39].

Among the limitations of the study, one can state the limited time of study due to lack of space and facilities for a longer period, hoped to be considered in future studies. As psychological factors like fear, anxiety, stress, depression, and lower self-esteem, activity and physical performance of older adults increase with aging and can increase the risk of falls and affect postural dynamic, it is recommended that the role of these agents be considered in future studies. Also, as sleep [40,41] and nap [42,43,44] are important for good performance, future studies could include the measurements of sleep and/or naps.

## 5. Conclusions

Overall, continuous physical activity can have a positive effect on motor function. It seems that designing aquatic training programs can be suitable especially for endomorphic older adults with inappropriate body types for land training, which can put a lot of pressure on joints, and who with less movement because of FOF. Moreover, given the unique effects of water on the control, physical, and emotional–nervous systems, it can bring about proper postural mobility away from FOF.

## Figures and Tables

**Figure 1 ijerph-16-04387-f001:**
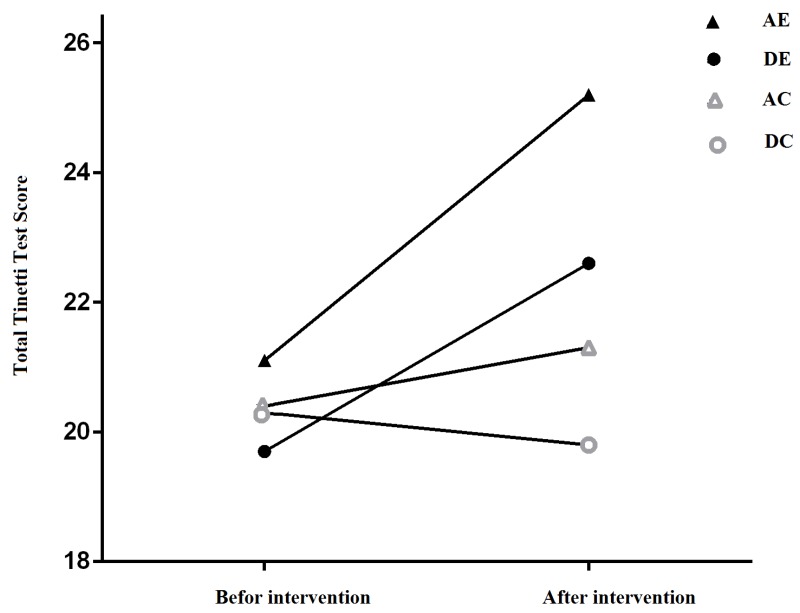
Total postural mobility score based on Tinetti test in the experimental groups before and after the intervention. AE: Aquatic Exercises, DE: Dry-land Exercises, AC: Aquatic Control, DC: Dry-land Control.

**Figure 2 ijerph-16-04387-f002:**
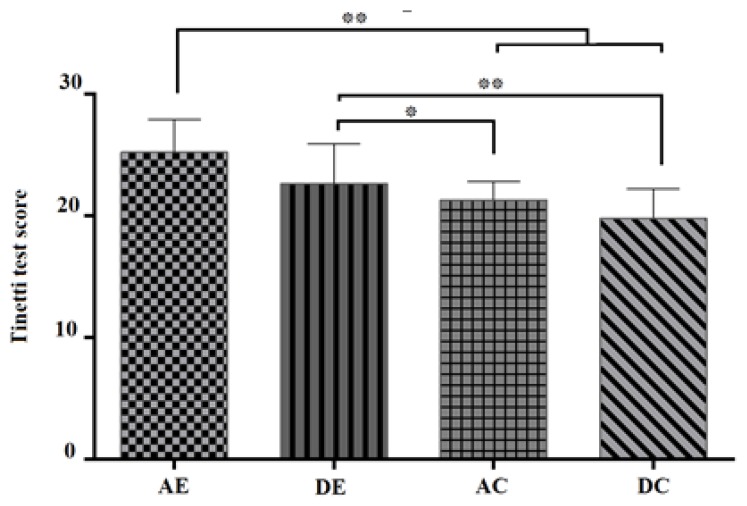
Bonferroni post-hoc test to compare the differences between groups; AE: Aquatic Exercises, DE: Dry-land Exercises, AC: Aquatic Control, DC: Dry-land Control. * *p* ≤ 0.05; ** *p* ≤ 0.01.

**Table 1 ijerph-16-04387-t001:** Heath-Carter computational method.

Body Type	Formula
Endomorphic	Endomorphy = −0.7182 + (0.1451X) − (0.00068X2) + (0.0000014X3) X = (H ÷ 170.118) × (Sum of subcutaneous fat in triceps; subscapularis muscle, supraspinatus muscle)
Mesomorphic	Mesomorphy = (0.858 × HB) + (0.601 × FB) + (0.188 × CAG) + (0.161 × CCG) − (H × 0.131) + 4.5
Ectomorphic	(a) If HWR was ≥ 40.74, then ectomorphy = (0.732 × HWR) − 28.58 (b) If 39.65 < HWR < 40.74, then ectomorphy = (0.463 × HWR) − 17.63 (c) If HWR ≤ 39.65, then ectomorphy = 0.1

HB: Distance between two lateral epicondylitis and the humerus (cm), FB: Distance between two lateral epicondylitis and femur (cm), CAG: Maximum arm circumference (cm), CCG: Maximum ankle circumference (cm), H: Height while standing (cm), HWR the cube root of body weight divided by height (cm/kg^3^).

**Table 2 ijerph-16-04387-t002:** Dry-land and Aquatic exercise protocol.

Step (Performance Time)	Protocol Summary
Warm-up (10 min)	Walking along side the pool and stretching exercises in different directions
The main set (30 min)	— walking forward and backward — walking backward while high stepping — marching forward and backward with knees bent — walking forward and backward with knees straight — side stepping without crossing the legs — side stepping with crossing the legs — heel-to-toe walking forward and backward — marching in place — standing partial squats — toe raises — heel raises — kicking in a diagonal — kicking in cardinal planes of motion — twisting [9]
Cooling down (10 min)	Walking, Breathing and flexibility exercises [24]

**Table 3 ijerph-16-04387-t003:** Individual Characteristics of the Subjects (Mean ± SD).

Group	AE (*n* = 10)	AC (*n* = 9)	DE (*n* = 10)	DC (*n* = 8)	Sig
Age (year)	64.61 ± 3.19	64.54 ± 3.74	63.91 ± 4.17	64.44 ± 3.54	0.219
Body height (cm)	162.4 ± 6.48	163.3 ± 5.10	165.1 ± 7.11	162.7 ± 5.81	0.077
Body weight (kg)	79.51 ± 3.33	81.35 ± 4.71	78.15 ± 6.55	80.47 ± 2.67	0.145
BMI (kg/m^2^)	31.57 ± 1.22	30.71 ± 0.38	30.60 ± 0.57	31.13 ± 0.88	0.579
WHR (Ratio)	0.92 ± 0.05	0.93 ± 0.04	0.92 ± 0.04	0.94 ± 0.05	0.098
BFP (%)	31.51 ± 4.12	33.11 ± 6.55	31.47 ± 4.34	30.32 ± 5.78	0.068
Endomorph (value)	8.7 ± 1.6	7.7 ± 1.8	8.5 ± 1.4	7.9 ± 1.4	0.135
Mesomorph (value)	6.1 ± 1.4	6.1 ± 1.7	6.0 ± 1.5	5.9 ± 1.9	0.644
Ectomorph (value)	0.3 ± 0.2	0.5 ± 0.4	0.5 ± 0.3	0.6 ± 0.2	0.805

WHR: Waist to Hip ratio, BFP: Body fat percentage, AE: Aquatic Exercises, DE: Dry-land Exercises, AC: Aquatic Control, DC: Dry-land Control.

**Table 4 ijerph-16-04387-t004:** Covariance test results for group comparison.

	SS	df	MS	F	Sig	Effect Size
Pretest	214.45	1	214.45	17.12	0.031	0.446
Groups	1685.14	3	561.71	25.38	0.000	0.741
Error	48.60	32	1.52	-	-	-

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
