# Peer review of "The Effect of Aquatic Exercise on Postural Mobility of Healthy Older Adults with Endomorphic Somatotype"

_ijerph, 2019, doi:10.3390/ijerph16224387_

Round 1

Reviewer 1 Report

Authors improved the manuscript considerably. Some parts still need to be reworked both because of poor clarity / grammar and for giving more information in the study context.  

I recommend that authors should describe exercise adherence, attendance respectively - i.e. how many exercise lessons the subjects visited out of 100 % of planned lessons. Also, as one of the control groups improved as well, it should be discussed - any idea why did this happen?

As for the language, it would be helpful to use shorter sentences so the reader is able to understand the text. Also, native English speaker should check once again the manuscript.

Notes: 

28: endomorphs elderly

37-38: provides an appropriate postural mobility without FOF in elderly - FOF was not measured in this study so the statement is not based on your results

51-52: Older adults try to reduce the risk of a fall by limiting their movement and activities to increase maintaining balance and postural mobility - possibly to rephrase to improve English and clarity of the text?

54: inaccurate cycle - why inaccurate?  perhaps vicious circle?

55: please add full term before first usage of abbreviation

56: Voluntary immobilization, movement diagnosis and body adaptations to the surrounding environment result all in both reduced physical activity and involuntary postural mobility - I do not understand how is movement diagnosis connected with reduced PA

60: anthropometric features (e.g., anthropometric measures such...) - not necessary to repeat words

65: As the article is dedicated to endomorph somatotype, it would be reasonable to describe at least this somatotype in the text.

79: factors

81: on the ground: do authors mean dry-land exercise or exercise on the floor in horizontal position? This should be clearly explained...

111: year,

Figure 1: befor

Figure 2 should be connected with text

238: Increase age

256-261: The sentence is very long and it is not easy to follow.

272- 276: Water is a fluid environment with medium density and viscosity that reduces the speed of movement, and due to this, when an individual is in the waist in the water, about 50% of his weight is lost, besides reducing gravity and weight loss in water, the probability of falling in the water is decreased [20]. This makes the older adults experience more mobility in water with more ROM. Thus, they experience more movement experiences [8, 20] - this part deserves to be rephrased

276: Moreover, the results showed that aquatic exercise improved water balance - is Tinetti testing water balance? It seems not. Then you cannot prove this...

Conclusion should be more focused on the results of the study.

Author Response

I recommend that authors should describe exercise adherence, attendance respectively - i.e. how many exercise lessons the subjects visited out of 100 % of planned lessons. Also, as one of the control groups improved as well, it should be discussed - any idea why did this happen?

Correction made as suggested in lines 119 to 121.

The slight difference in the water control group is probably due to the effective innate characteristics of   water that even untrained persons who go into the water would have improvement because of weight loss, sweating and hydrostatic pressure with water turbulence.

As for the language, it would be helpful to use shorter sentences so the reader is able to understand the text. Also, native English speaker should check once again the manuscript.

Revision made as suggested.

28: endomorphs elderly

Corrected: line 26

37-38: provides an appropriate postural mobility without FOF in elderly - FOF was not measured in this study so the statement is not based on your results

Corrected as follows: "both the balance and gait provides an appropriate postural mobility in older adults".

Please see changes made in the text.

51-52: Older adults try to reduce the risk of a fall by limiting their movement and activities to increase maintaining balance and postural mobility - possibly to rephrase to improve English and clarity of the text?

Line 53: Reworded as follows: “Older adults try to reduce the risk of a fall by limiting their movement and activities so that they would   increase balance and postural mobility”.

54: inaccurate cycle - why inaccurate?  perhaps vicious circle?

Corrected.

55: please add full term before first usage of abbreviation

Corrected “Fear Of Falling (FOF)”.

56: Voluntary immobilization, movement diagnosis and body adaptations to the surrounding environment result all in both reduced physical activity and involuntary postural mobility - I do not understand how is movement diagnosis connected with reduced PA

Line 57-60: Changed as follows:

“Movement errors have been well documented as the guidance for motor skill acquisition. Similarly, the failure to make postural movement errors may lead to a loss of skill in performing postural tasks. Thus, older adults' limited mobility may cause a progressive reduction in their postural capacity by their failure to produce errors while practicing these skills. They also increase the postural instability and the fear of falling. This cycle frequently happens until it makes the older adults severely homebound.”

60: anthropometric features (e.g., anthropometric measures such...) - not necessary to repeat words

Deleted.

65: As the article is dedicated to endomorph somatotype, it would be reasonable to describe at least this somatotype in the text.

Line 66: the following sentence was added:

“endomorph (which expresses body fat content)”.

79: factors

Done.

81: on the ground:do authors mean dry-land exercise or exercise on the floor in horizontal position? This should be clearly explained...

We meant exercise in lands. It is a specialized word in swimming area.

111: year,

Line 110: corrected.

Figure 2 should be connected with text

Lie 214: Correction made as suggested.

238: Increase age

Corrected.

256-261: The sentence is very long and it is not easy to follow.

Line 258-259: Changed as follows:

“Obese people are 20% or more above heavier than the ideal level of normal body weight Obese people are 20% or more above …..”

272- 276: Water is a fluid environment with medium density and viscosity that reduces the speed of movement, and due to this, when an individual is in the waist in the water, about 50% of his weight is lost, besides reducing gravity and weight loss in water, the probability of falling in the water is decreased [20]. This makes the older adults experience more mobility in water with more ROM. Thus, they experience more movement experiences [8, 20] - this part deserves to be rephrased

Line 276-278: Rephrased.

276: Moreover, the results showed that aquatic exercise improved water balance - is Tinetti testing water balance? It seems not. Then you cannot prove this...

Corrected as follows:

“Moreover, the results showed that aquatic exercise improved balance due to neural adaptations induced by exercise in water, such as the use of neural efficient reorganization of cortical somatosensory.”

Conclusion should be more focused on the results of the study.

The main results are that:

“Due to the improvement in the results of the Titanic (Balance) test in the drought and water training groups, it is better for the elderly to use the water exercise because of the safer environment and more properties of water to land (weight loss, hydrostatic pressure, turbulence, resistance).”

Reviewer 2 Report

It is an interesting study and I believe that it can be validated as a strengthening element of the use of aquatic therapy in patients with certain age and physical characteristics, which influence the loss of balance.
In general, it is correct, but in my opinion, improvements should be made in the presentation of the text.
- It would be appreciated if a figure were presented showing the patient selection process, to make it more visual and clarifying.
- Below the figure that shows the results obtained in the Tinetti scale, the name corresponding to the acronyms indicated must be indicated.
- On line 263: what does AEA (2017) mean?
- In conclusion: lines 286, 287 and 288 are not indicated in this section. I think it should be included in discussion.

Author Response

It is an interesting study and I believe that it can be validated as a strengthening element of the use of aquatic therapy in patients with certain age and physical characteristics, which influence the loss of balance. In general, it is correct, but in my opinion, improvements should be made in the presentation of the text.

- It would be appreciated if a figure were presented showing the patient selection process, to make it more visual and clarifying. It's well explained now in page 3 line 105-109.

Below the figure that shows the results obtained in the Tinetti scale, the name corresponding to the acronyms indicated must be indicated.

Correction made as suggested.

- On line 263: what does AEA (2017) mean?

It represent the Aquatic Exercise Association (AEA) and it is now explained in line 183.

In conclusion: lines 286, 287 and 288 are not indicated in this section. I think it should be included in discussion.

Correction made as suggested.

Reviewer 3 Report

The idea is clear and the manuscript is well written. However, I have minor comments prior to acceptance.

Line 32: P-value should be written in italic form. The same should be done in the whole text. Line 88: "F" in favorable should be a capital letter.

Author Response

The idea is clear and the manuscript is well written. However, I have minor comments prior to acceptance.

Line 32: P-value should be written in italic form. The same should be done in the whole text. Line 88: "F" in favorable should be a capital letter.

The following values are in italics and F is large in text with blue highlighting